# Complete Genomic Characterisation and Mutation Patterns of Iraqi SARS-CoV-2 Isolates

**DOI:** 10.3390/diagnostics13010008

**Published:** 2022-12-21

**Authors:** Jivan Qasim Ahmed, Sazan Qadir Maulud

**Affiliations:** 1Department of Pathology and Microbiology, University of Duhok, Duhok 42001, Iraq; 2Department of Biology, College of Education, Salahaddin University-Erbil, Erbil 44001, Iraq

**Keywords:** SARS-CoV-2 evolution, mutations, COVID-19, variants, clades, lineages

## Abstract

This study was performed for molecular characterisation of the SARS-CoV-2 strains in Iraq and reveal their variants, lineages, clades, and mutation patterns. A total of 912 Iraqi sequences were retrieved from GISAID, which had been submitted from the beginning of the SARS-CoV-2 pandemic to 26 September 2022, along with 12 samples that were collected during the third and fifth waves of the SARS-CoV-2 pandemic. Next-generation sequencing was performed using an Illumina MiSeq system, and phylogenetic analysis was performed for all the Iraqi sequences retrieved from GISAID. Three established global platforms GISAID, Nextstrain, and PANGO were used for the classification of isolates into distinct clades, variants, and lineages. Analysis of the isolates of this study showed that all the sequences from the third wave were clustered in the GK clades and the 21J (Delta) clade according to the GISAID and Nextclade systems, while the PANGO system revealed that six sequences were B.1.617.2 and four sequences were of the AY.33 lineage. Furthermore, the latest e wave in the summer of 2022 was due to thpredominance of the BA.5.2 lineage of the 22B (Omicron) clade in Iraq. Our study revealed patterns of circulation and dominance of SARS-CoV-2 clades and their lineages in the subsequent pandemic waves in the country.

## 1. Introduction

The novel coronavirus disease (COVID-19) is caused by a severe acute respiratory syndrome coronavirus 2 virus (SARS-CoV-2), which initially emerged in China in late December 2019 [1,2]. Later on, as a result of the fast spread of the virus to many other countries around the globe, the state of pandemic was declared by the World Health Organisation (WHO) in March 2020 [3]. As of October 2022, there had been more than 600 million cases of infection and more than 6 million deaths caused by SARS-CoV-2, according to the weekly situation reports of the WHO. In Iraq, from 3 January 2020 to 23 December 2021, there were more than 2 million confirmed cases and more than 25 thousand deaths due to SARS-CoV-2, as reported by the WHO [4]. Meanwhile, researchers are trying to understand the evolution process and classification of the virus into evolutionary groups that may have distinct phenotypes using many tools, particularly genomic sequencing. In the meantime, more than 13 million SARS-CoV-2 whole genome sequences have been submitted to the Global Initiative on Sharing All Influenza Data (GISAID) database [5].

SARS-CoV-2 is a member of beta-coronaviruses that is included in the Coronaviridae family, Coronavirinae subfamily [1]. Similarly to other coronaviruses, SARS-CoV-2 is an enveloped virus, and the genomic RNA is single-stranded, with positive-sense polarity [6]. The SARS-CoV-2 genome consists of around 29 kb nucleotides that are arranged to untranslated regions (UTR) at both the 5′ and 3′ ends, open reading frame 1ab (ORF1ab), structural proteins, and many other accessory proteins [7]. The ORF1ab region constitutes over two-thirds of the genomic RNA, which encodes 16 nonstructural proteins. The structural proteins that are encoded in the genomic RNA include the spike protein (S), the envelope protein (E), the membrane protein (M), and the nucleocapsid phosphoprotein (N). In addition, there are six more proteins encoded at the downstream region of the genomic RNA: ORF3a protein, ORF6 protein, ORF7a protein, ORF7b protein, ORF8 protein, and ORF10 protein [8,9]. The spike protein that protrudes from the viral particle is an essential viral protein for binding to the human angiotensin-converting enzyme 2 (ACE2) receptor that enables viral entry and infection, defines tissue tropism and transmission capacity. SARS-CoV-2 has the nsp14 exonuclease, which significantly reduces the mutation rate to a level similar to that of DNA viruses and consequently allows it to develop some of the largest genomes among all RNA viruses [10,11,12]. The mutations and evolution of the SARS-CoV-2 genome could increase viral transmission [13], host receptor binding affinity, host tropism, and increasing viral pathogenicity [7].

Mutations could occur in the genomic RNA of the SARS-CoV-2 virus to evade the host’s immune response. Such mutations could result in the emergence of new variants with increased infectivity, transmission, and pathogenicity [14]. As a result of consecutive waves of the pandemic around the globe [15], there is a need to monitor the virus’ evolution and analyse any potential mutations that could alter the pathogenesis and virulence of the virus [16]. Countries around the world are monitoring the virus’ evolution in local communities through whole genome sequencing of SARS-CoV-2. In Iraq, sequencing of the virus has been in limited numbers since the first emergence of infection cases, and there are only 912 complete genome sequences that have been submitted to the GISAID database since the beginning of the pandemic.

The purpose of this study was to perform whole genome sequencing of the SARS-CoV-2 virus isolates circulating in the Duhok Governorate of Iraq, as well as the identification of dominant variants, clades, and lineages and to reveal the potential mutation patterns in the sequences of SARS-CoV-2 from the third and fifth waves of the pandemic that peaked in the summer periods of 2021 and 2022.

## 2. Materials and Methods

### 2.1. Samples

From the beginning of the pandemic, the Duhok Central public health laboratory was designated the general public sector laboratory for molecular testing for COVID-19. Nasopharyngeal and oropharyngeal swabs from patients in different parts of the Duhok Governorate were collected in a viral transport medium according to the guidelines of the WHO. At the time of the third wave of the SARS-CoV-2 pandemic in the summer of 2021, when the country was suffering from the strongest wave since the beginning of the pandemic, 60 previously tested positive samples were provided to us by the Duhok Central public health laboratory upon request. Later, two representative samples from the latest pandemic wave were also included in the analysis. These samples were collected from different parts of the Duhok Governorate from 1 July to 30 July 2021, and two other samples were collected in August and September 2022 for the purpose of routine COVID-19 testing. The SARS-CoV-2-positive samples went through variant secreening using a PowerChek™ SARS-CoV-2 S-gene Mutation Detection Kit (ver. 03) (Kogene), and representative samples from each wave were selected for sequencing. Quantitative real-time PCR was performed for quality control purposes, and 12 samples of a low Ct (< 20) value were randomly selected for next-generation sequencing. This study was approved by the ethics committee of the Directorate General of Health in Duhok (18502022-3-22). All the participants, including parents of the minors, provided written consent for participating in this study.

### 2.2. Viral RNA Extraction and Real-Time PCR

SARS-CoV-2 detection and confirmation were carried out using a QIAprep & Viral RNA UM Kit (Qiagen), a both RNA extraction and real-time PCR-based virus detection system integrated into one single kit. The samples with a Ct value lower than 20 were selected for whole genome sequencing. RNA extraction of the selected samples from the viral transport medium was carried out using a QIAamp Viral RNA Mini Kit (Qiagen) and transported in dry ice conditions to Intergen (Ankara, Turkey) for next-generation sequencing.

### 2.3. SARS-CoV-2 Next-Generation Sequencing (NGS)

The samples were commercially sequenced at the Intergen Genetics and Rare Diseases Diagnosis Research and Application Centre (Ankara, Turkey). Briefly, after receiving the RNA samples at the sequencing centre, they were tested again for confirmation and RNA integrity. The RNA samples were reverse-transcribed with an Ipsogen RT Kit using random nanomers as primers (Qiagen, Germany). On the obtained cDNA samples, amplicons including the target loci were amplified via polymerase chain reaction (PCR) using in-house-designed primers. PCR products were visualised on a 2% agarose gel stained with ethidium bromide under ultraviolet light. An Illumina sample preparation kit was used for tagmentation and indexing of the individual samples. Illumina MiSeq devices and equipment (Illumina, San Diego, CA, USA) were utilised to perform next-generation sequencing following the manufacturer’s instructions. The sequencing data were evaluated using IGV 2.8.9 (Broad Institute).

### 2.4. Bioinformatics Analysis

The sequencing reads were aligned and assembled using reference sequence NC_045512.2 and alignment algorithm BWA-MEM [17]. At the same time, variant calling and mutation identification against the reference genome (NC_045512.2) were performed using Lofreq (version 2) [18]. Additionally, the assembled sequences’ annotation was performed using Annovar as previously described [19]. All sequence reads were subjected to quality control before acceptance. Sequences of > 99% genome sequencing coverage and gap length of fewer than 30 bps were selected for additional investigation. Finally, a complete set of 12 samples of complete genome sequences passed quality control successfully and was subsequently submitted to the GISAID database, where the following sequence accession numbers were assigned: EPI_ISL_5804791, EPI_ISL_5804792, EPI_ISL_5804793, EPI_ISL_5804794, EPI_ISL_5804795, EPI_ISL_5804796, EPI_ISL_5804797, EPI_ISL_5804798, EPI_ISL_5804799, EPI_ISL_5804800, EPI_ISL_15085309, EPI_ISL_15085310.

### 2.5. Lineage and Phylogenetic Analysis

The 12 sequenced samples and 912 complete genome sequences of Iraqi isolates were retrieved from the GISAID database from the start of the pandemic to 26 Septemeber 2022. The SARS-CoV-2 sequences of the Iraqi isolates were subjected to lineage identification by the Pangolin system (v3.1.14) [20]. Furthermore, the clades were identified using the Nextclade sequence analysis and the GISAID database platform tools [5,21].

As well, a dataset was prepared using representative Iraqi isolates from subsequent SARS-CoV-2 pandemic waves. Furthermore, some closely related global sequences were retrieved from the National Centre for Biotechnology Information (NCBI) through blast searching of Duhok sequences. In addition, whole genome sequences of SARS-CoV-2 from the neighbouring countries (Turkey, Saudi Arabia, Jordan, and Iran) reported from the beginning of May 2021 to the end of September 2022were retrieved from the GISAID database. The number of selected sequences was primarily dependent on the availability and quality of sequences; the sequences with extended (N) sites were excluded from the analysis. The phylogeny was constructed using the maximum likelihood method and the general time reversible model as implemented in Molecular Evolutionary Genetics Analysis (MEGA) version 11 [22].

## 3. Results

This study aimed to characterise 12 samples of the SARS-CoV-2 whole genome sequences isolated from the Duhok Governorate and another 912 Iraqi sequences on GISAID from the beginning of the pandemic to September 2022. In the meantime, three established global platforms were extensively used globally to classify and identify SARS-CoV-2 isolates into distinct clades, variants, and lineages. The platforms are GISAID, Nextstrain, and PANGO [20,23,24,25,26]. These systems have come up with different naming systems and brought about a need in a universal and uniformed platform.

Using the GISAID/Nextrain system of isolate classification, the isolates of the third wave were clustered in the GK clade (Delta variant), while sequences of the fifth wave were GRA (Omicron variant). In contrast, the other Iraqi isolates were assigned to various clades through subsequent pandemic waves, as illustrated in Figure 1. The second nomenclature system used was the Nextclade tool of the Nextrain platform; the resulting clade for the Duhok isolates in this study were 21J (Delta variant of concern) for the third wave and 22K (Omicron) for the fifth wave, which was the leading isolate of the fifth wave. In comparison, the other Iraqi isolates were assigned to several different clades, including 20A, 20B, 20C, 20E, 20H (Beta), 20I (Alpha variant, V1), 19A, and 21J (Delta variant), and 21K (Omicron). The most common clades discovered in the Iraqi isolates were 21J (Delta variant; 53%) and 21K (Omicron; 29%) (Figure 2).

For further classification into lineages and sublineages, the PANGO system was used. As a result, the 10 sequnces of the third wave were as follows: six sequences were B.1.617.2 and four sequences were identified as the AY.33 lineage, which are sublineages of the Delta variant of concern within the GK clade. On the other hand, the two other samples from the latest pandemic wave were BA.5.2 (Omicron variant), identified in the country for the first time. At the same time, the other Iraqi isolates were assigned to 25 different circulating lineages, including B.1.177, B.1.1, B.1.36.1, B.1.36. B, B1, B.1.351, B.1.1.7, AY.122, AY.33, B.1.617.2, AY.46, AY.4, AY.43, AY.106, AY.121, AY.126, AY.65, AY.121, AY.5, AY.98, BA.1.1, BA.1, BA.2, and BA.5.2 (Figure 3). Furthermore, the predominantly circulating lineage was found to be AY.33 (25%), which was identified in most Iraqi sequences, considering the limited sampling procedure (Figure 3).

### 3.1. Phylogenetic Analysis of the Third Wave Isolates and Other Iraqi Sequences

Whole genome sequences of the Duhok isolates in this study representing the third wave of the pandemic in the country and the other Iraqi isolates were analysed. A phylogenetic tree was constructed using the Nextclade web-based application that uses global sequences from the GISAID database for phylogenetic analysis. The 10 representative sequences from the third pandemic wave were clustered as 21J (Delta variant), which was detected in three other sequences at the end of April and in June 2021 (Figure 4). Furthermore, the two representative isolates from the fifth wave were clustered as 22B (Omicron), which had not been detected in the country before. Furthermore, a phylogenetic tree was constructed using the GISAID database global sequences filtering to Iraqi sequences to show the timing of variant appearance in the country (Figure 5).

Concerning the phylogeny constructed using the second dataset of closely related sequences from all over the world and the neighbouring countries, the predominantly circulating GK clade in Duhok was also identified in Jordan, Saudi Arabia, Kuwait, and Turkey during the active third wave of the pandemic in the country (Figure 6). The Duhok isolates were found to be closely related to the sequences from the USA, the United Kingdom, Germany, Switzerland, and Japan. On the other hand, the fifth wave isolates were found to be close to sequences from the United States and Slovakia (Figure 6).

### 3.2. Mutations in SARS-CoV2 Genomes of the Duhok Isolates during the Third Wave of the Pandemic in Iraq

Compared to the genomic reference sequence (NC_045512), the Duhok SARS-CoV-2 sequences of the third wave showed a total of 102 different mutations. Of the 102 mutations, there were 63 nonsynonymous mutations, 31 synonymous mutations, two deletions, and six mutations in the untranslated region identified in 10 isolates of the study (Appendix A). These mutations resulted in 63 different amino acid variants; the majority of them were found on the spike protein (22), nsp3 (17), and nsp12 (10). In contrast, the sequences of the fifth wave (Omicron variant) showed a total of 80 different mutations. Of those mutations, there were 53 nonsynonymous, 20 synonymous mutations, four deletions, and three mutations in the untranslated region (Appendix A). These mutations resulted in 53 different amino acid variants; the majority of them were found on the spike protein.

In terms of their distribution by genomic regions and genes for the Delta variants of this study, 53 mutations were found in ORF1ab (51.9%) alone, and subgenomic region nsp12 (RNA-dependent RNA polymerase) was found to be the hotspot of mutations (Table 1). The second most mutated gene was the S gene, where 22 (21.5%) mutations were detected, followed by seven mutations in the N gene (6.8%), four (3.9%) in the ORF7a gene, four (3.9%) in the ORF3a gene, two (1.9%) in the M gene, and one (0.9%) mutation in each of the ORF6, ORF7b, ORF8, and ORF10 genes. Nevertheless, no mutations were detected in the E gene of the Delta isolates of this study. On the other hand, the Omicron variant from the fifth wave showed the spike protein gene to be the most mutated region with 32 mutations (39%), followed by the OR1ab region with 27 (33%) different mutations (Figure 7 Table 1, Appendix A).

Regarding the incidence and frequency of every single mutation, a total of 23 conserved mutations were found in all the 10 sequences of the Duhok isolates. Of these mutations, 20 were on the coding regions, three—on the upstream and downstream regions. Furthermore, 18 of these conserved mutations were nonsynonymous mutations, two were deletions, and one was a synonymous mutation. The conserved protein variants in all the 10 isolates were as follows: nsp3-F106F, nsp12-P323L, nsp12-G671S, nsp13-P5401L, S-T19R, S-G142D, S-156_158del, S-L452R, S-T478K, S-D614G, S-P681R, S-D950N, ORF3A-S26L, M-I82T, ORF7a-V82A, ORF7a-T120I, ORF8-119_120del, N-D63G, N-R203M, and N-D377Y (Appendix A). Mutations nsp3-S915R (0.02%), nsp12-K103R (0.06%), nsp15-D6734Y (0.04%), and N-L221F (0.03%) of the Duhok sequences were found to be very rarely reported, while the incidence of the variants S-D80Y, ORF6-P57L, and N-T135I was not found to be reported on the GISAID database. In contrast, the highly mutated Omicron variants of the fifth wave showed mutations that were not detected before in the Delta variants: S-A27S, S-V213G, S-G339D, S-S371F, S-S373P, S-S375F, S-T376A, S-D405N, S-R408S, S-K417N, S-N440K, S-S477N, S-E484A, S-F486V, S-Q498R, S-N501Y, S-Y505H, S-H655Y, S-N679K, S-P681H, S-N764K, S-D796Y, S-Q954H, S-N969K ORF1a-S135R, ORF1a-T842I, ORF1a-G1307S, ORF1a-L3027F, ORF1a-T3090I, ORF1a-P3395H, ORF1a-L3711F, ORF1b-P314L, ORF1b-T1050N, ORF1b-R1315C, ORF1b-I1566V, ORF1b-T2163I, ORF3a-T223I, ORF3a-E239D, ORF9b-P10S, ORF9b-D16G, E-T9I, M-D3N, M-Q19E, M-A63T, N-P13L, N-R203K, N-G204R, and N-S413R. Furthermore, some mutations such as S-T19R, S-G142D, S-L452R, S-T478K, S:D614G, and ORF1b-T3255I were found to be conserved in the Omicron variant from the Delta variant (Figure 6).

## 4. Discussion

This study showed molecular characterisation and mutation patterns of the circulating SARS-CoV-2 strains in the Duhok Governorate throughout the third and fifth waves of the pandemic when the daily cases reached their highest peak since the beginning of the pandemic. In the beginning of the summer of 2021, the country was going through the worst wave of the pandemic. Therefore, it was necessary to conduct a study on monitoring the evolution of the virus in the country. Before, no study had been conducted to monitor and identify the top circulating strains and variants in the country. Furthermore, as of September 2022, only 924 whole genome sequences from Iraq had been submitted to the GISAID database, including the samples of this study, and no samples had been sequenced since February 2022; this created a considerable gap in monitoring the emerging variants and identification of the circulating SARS-CoV-2 variants in the country.

In terms of the clades and lineages dominating the third wave, our study reported that the sequenced samples of the third wave belonged to the Delta variant of concern (clade 21J); this variant had been detected before in the Dhi Qar and Erbil Governorates at the end of April and in June 2021, respectively. Afterwards, the Delta variant of concern started to spread over the country and dominate the circulating variants: of all the 924 sequenced samples, 491 (53%) isolates were found to be of the Delta variant (Figure 2). The Delta variant that emerged in India in late 2020 and was discovered to have increased transmissibility and infectivity [27], a shorter incubation period [28], as well as the potential to neutralise antibody evasion compared to the other variants of SARS-CoV-2 [29]. These features of the Delta variant resulted in it becoming the dominant circulating variant and spreading more rapidly in the country. As a result, the emergence of the Delta variant had caused a sharp increase in daily cases by July 2021 in the country, which was the most substantial wave to hit the country since the beginning of the pandemic, according to the data from both the health ministries of the Kurdistan Region and the federal government in Baghdad (Figure 8). At the same time, the samples of the latest fifth wave belonged to the Omicron variant of concern (clade 22B) that had never been identified or detected in the country.

On the other hand, the GISAID clade prevalent in the country during the first wave of the pandemic was predominantly GR 141 (15%) for the samples sequenced from the beginning of the pandemic to the emergence of the Delta variant in April 2020 (Figure 1). Afterwards, GK clade 484 (52%) (Delta variant) dominated the circulating strains, leading to the third wave of the pandemic in the country until the emergence of the Omicron variant in January 2022. Since the first identification of the Omicron variant in January 2022 [30], its clade GRA has been the dominant clade in the country. Interestingly, during the daily cases surge in the summer of 2021, the GK clade dominated the circulating strains identified in all the isolates of this study at the time of the third wave of the COVID-19 pandemic, followed by the GRA clade (Figure 1). These data are in line with findings from many other studies that concluded that more transmissible variants can replace the original variants [31,32]. It should be noted that many lineages of those previously explained clades were circulating in the country, according to the sequenced samples, and B.1.1.7 was the dominant lineage for the period from January to July 2021.

Nonetheless, during the July mass wave, B.1.617.2 and AY.33 of the Delta variant were the predominant circulating lineages based on the sequences of this study and other sequences from Iraq (Figure 3). The lineages BA.1.1, BA.1, and BA.2 of the Omicron variant were the most prevalent from January to August 2022. This study identified the BA.5.2 lineage of the Omicron variant for the first time in the country, which was in accordance with the rise of the fifth wave (Figure 3).

High rates of mutations occur in the genomes of RNA viruses; they are usually used to overcome a changing environment [33], evade the host’s immune response and the effect of antiviral drugs [34]. This coronavirus has been shown to have a unique proofreading mechanism during genomic replication compared to other RNA viruses [10,11,12,33] through employing RNA-dependent RNA polymerase, nsp7, nsp8 [35], and exonuclease activity (nsp14) [10,11,12]. Therefore, the mutation rate is slow, estimated at around 1–2 point mutations in SARS-CoV-2 per month [6]. One study analysing and monitoring the SARS-CoV-2 evolution recorded 3–22 single mutations per genome [14], another study—7–17 mutations per genome [6]. This number of mutations nearly doubled in the Duhok isolates from the third wave (n = 39–47 mutations per genome) and doubled again in the isolates of the fifth wave (n = 75 mutation per genome) compared to our results; this was probably a result of accumulating mutations in the genome of SARS-CoV-2 (Figure 8). Of the 102 mutations found in the Duhok isolates, 22 were found in all the SARS-CoV-2 isolates of this study. It was concluded in an earlier study that nonsynonymous mutations are more frequent than the synonymous ones; therefore, it is believed that these mutations are a result of positive selection by the SARS-CoV-2 virus [7]. These findings are in line with the results of this study, where nonsynonymous mutations were 61.7% and 53% versus 30.3% and 20% for synonymous mutations in the sequences from the third and fifth waves, respectively.

The results of Duhok sequence analysis revealed that the cytosine (C) to thymine (T) mutation was the most common, which is in line with the previous reports [9,36,37]. In terms of genomic regions, ORF1ab constituted two thirds of the genome, but it accumulates 51% of the mutations for the Delta variant isolates, which agrees with the outcomes of the previous reports [9,37], and the results for the Omicron variants totally disagree as the spike protein was found to be the most mutated (39%; n = 32).

Among the nonstructural proteins of the ORF1ab gene, nsp3 was the most mutated protein in the third-wave sequences, constituting 32% of all the mutations in the ORF1ab region, and only had two mutations in the fifth-wave isolates. Mutations 3037 C > T, 4181 G > T, and 6402 C > T were the most commonly detected ones in nsp3 of the third-wave isolates, resulting in variants F106F, A488S, and P1228L, respectively; these results are in line with a previous report [6] where nsp3 of the Omicron variant did not acquire any mutations from the Delta variant (Figure 6). It is belived that the nsp3 protein, along with nsp4 and nsp6, is involved in double-membrane activation, which facilitates the replication and transcription process [38] and is involved in the evolution process responding to the selective pressure on the virus [39]. In contrast, variations in nsp12 were 18.8% of the ORF1ab mutations for the Delta variant, and there was only one mutation in the Omicron isolates. The nsp12 protein encodes RdRp, functions to replicate the viral genome, and has a vital role in the proofreading process [35].

The viral spike protein functions as a glycoprotein for infection initiation that recognises and binds to the angiotensin-converting enzyme 2 receptors on the target cells [40]. As a result of viral spillover from animals to its new host, the virus has undergone several mutations in the spike protein. Consequently, the extensively detected variant D614G in the spike gene, which had been previously related to the increasing infection and transmission rates of SARS-CoV-2 around the globe [41] due to its high binding affinity to the target receptor, is a good example. Based on our analysis, variant D614G was highly circulating in Duhok and probably in other parts of Iraq; this mutation could be the leading cause of the increased transmission and infection rates. Similarly, many other mutations were detected in all the Duhok sequences from the third wave, including 21,618 C > G, G21987 G > A, 22,917 T > G, 22,995 C > A, 23,403 A > G, 23,604 C > G, 24,410 G > A, and 25,469 C > T, that had the following amino acid variants T19R, G142D, L452R, T478K, D614G, P681R, D950N, and S26L, respectively. In addition, we detected six nucleotide deletions at position 22,029 on the spike gene of the virus. On the other hand, the spike gene was found to be a mutation hotspot in the sequences of the BA.5 lineage of the Omicron 22B clade.

The third most mutated viral protein was the nucleocapsid protein in both groups of sequences; this protein has a vital role in the transcription and replication of the viral RNA [42]. Interestingly, antibodies against this protein were detected during the infection [43]. This could be a potential target protein for vaccine development [44]. The previous reports of the global sequence analysis showed highly dominant R203K and G204R variants on the N gene worldwide at the same time as three nucleotides at positions 28881–28883 (GGG to AAC) [6,9,45,46,47]. While our sequences showed the disappearance of these three mutations that had been detected in most of the Iraqi sequences of the earlier consecutive pandemic waves in the country, the mutation was replaced by 28,881 G > T with amino acid substitution R203M in the sequences. Furthermore, variant D63G also had the dominant mutations in the third-wave sequences, and the D63G, R203M, and D377Y variants were detected among the Delta variants in India [48].

Accessory proteins ORF6, ORF7, and ORF8 are the main viral components for the suppression of the innate immune system, IFN expression, and signalling pathways [49]. Mutations on these proteins were less frequent in our study sequences, which agrees with data from other reports [6]. Similarly, mutations on ORF3a were less frequent compared to other genomic regions, and only mutation 25,469 C > T with amino acid substitution S26L was recorded in the third-wave sequences of this study and replaced by T223 and E239D in the BA.5 lineage from the fifth wave.

Mutation 241 C > T of the 5′-UTR region is the most commonly detected one worldwide [50]. This mutation was detected in all the Duhok sequences from both waves. The reports on the analysis of viral genomes suggested that mutations on untranslated regions could affect packaging and viral RNA folding. Besides, this mutation was discovered to accompany three variants, ORF1ab-P4715L, nsp12-P323L, and S-D614G [47]. In contrast, in our study, mutation P323L on the RdRp protein was detected only in one sequence.

This study had some limitations: the sample size was not sufficient, and it was not practical to perform whole genome sequencing for every sample. Therefore, the selection bias was decreased by means of blind random selection of 12 samples for complete genome sequence variation analysis, but random error could not be eliminated. However, the size of the selected samples was limited; this was the first study to perform whole genome sequencing of the SARS-CoV-2 strains from the Duhok Governorate. Therefore, these significant findings can be used as the starting point for upcoming epidemiological and molecular studies and the clue for vaccine development and diagnostic approaches [44,51]. Furthermore, sequencing and monitoring of the SARS-CoV-2 virus and many other pathogens in Iraq is very slow due to the lack of expertise and sequencing facilities and funding. Therefore, the government and research laboratories need to perform large-scale and more in-depth sampling and sequencing to monitor the virus’ evolution in local communities and share such data on the global database for further studies and surveillance purposes.

## 5. Conclusions

In conclusion, the results of our study revealed that the third wave in the summer of 2021 in the Duhok Governorate was caused by the emergence of the Delta variant of SARS-CoV-2 of clade GK. The rapid transmissibility and infectivity of the Delta variant were behind an abrupt increase in daily cases in the province. In its turn, the latest wave of the pandemic was identified to have been caused by the BA.5 lineage of the Omicron variant (22B) that had never been detected in the country. The emergence of these variants was recorded in many vaccinated patients in the province; this brought about the need to adopt new vaccines to include the emerging variants of concern. Therefore, having knowledge of mutation patterns over time and other molecular characteristics of the SARS-CoV-2 virus would be valuable for clinical considerations, planning public health and control measures, and therapeutic drug and vaccine development. Further and continuous studies are required for surveillance and molecular characterisation of the circulating strains in the country. The public sector and government health agencies should utilise whole genome sequencing strategies at least on a monthly basis to track the virus’ evolution and emergence of any potential variants that could have public health importance.

## Figures and Tables

**Figure 1 diagnostics-13-00008-f001:**
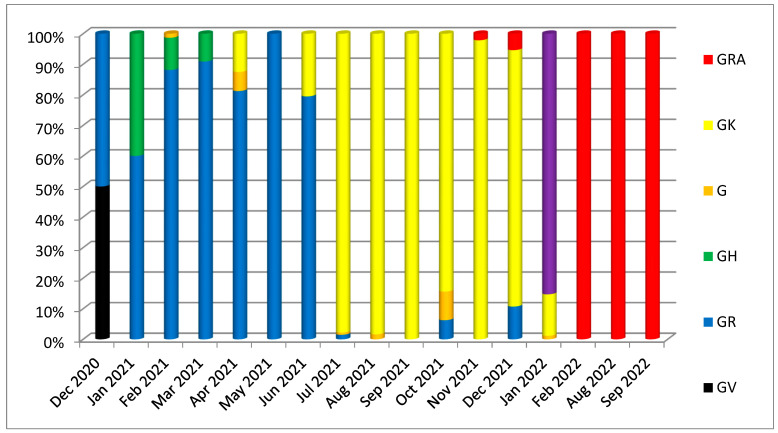
Circulation of SARS-CoV-2 clades in Iraq during five pandemic waves assigned using the GISAID clades.

**Figure 2 diagnostics-13-00008-f002:**
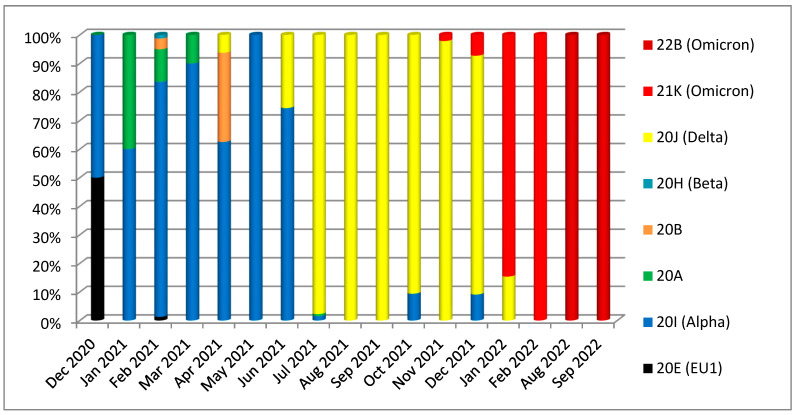
Circulation of SARS-CoV-2 clades (variants) in Iraq during five pandemic waves using the Nextclade system.

**Figure 3 diagnostics-13-00008-f003:**
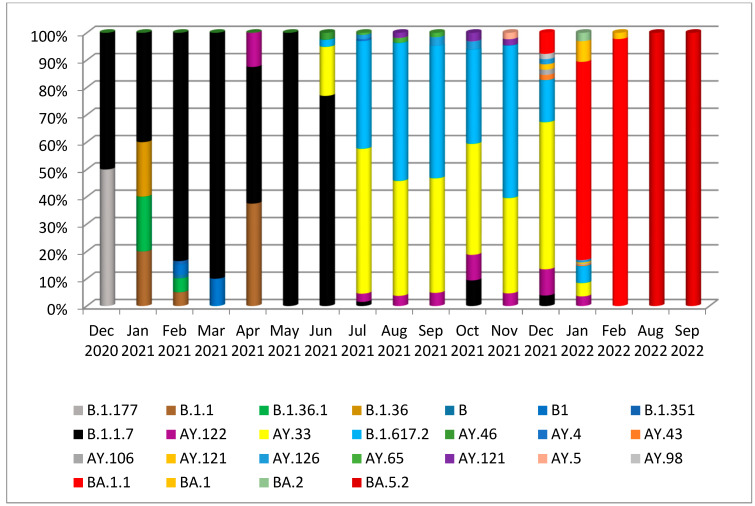
Circulation of SARS-CoV-2 lineages in Iraq during five pandemic waves.

**Figure 4 diagnostics-13-00008-f004:**
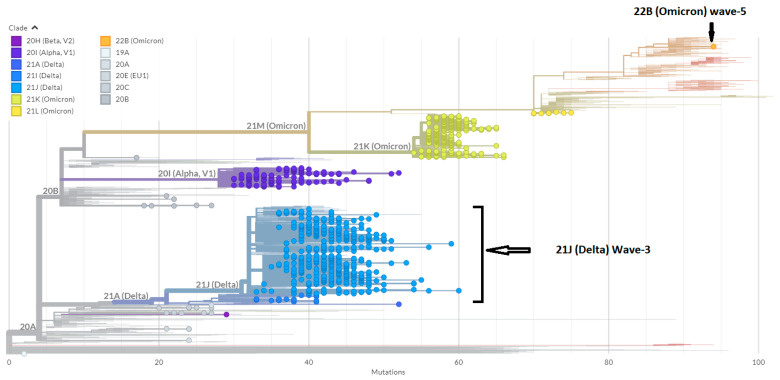
Phylogenetic tree of whole genome sequences from the third and fifth waves’ samples and the other Iraqi SARS-CoV-2 isolates from the beginning of the pandemic with all global isolates demonstrating the position and clustering of the named clades (Nextrain nomenclature). The tree was constructed without any restrictions using the Nextclade web service (https://clades.nextstrain.org/ accessed on 1 November 2022).

**Figure 5 diagnostics-13-00008-f005:**
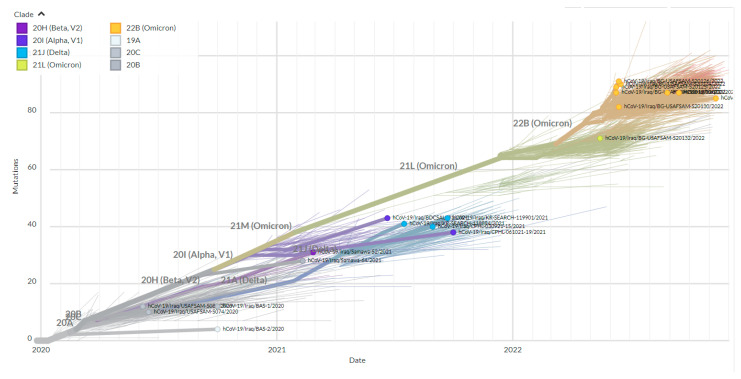
Nextrain phylogeny shows the time of identification of each clade in the Iraqi isolates.

**Figure 6 diagnostics-13-00008-f006:**
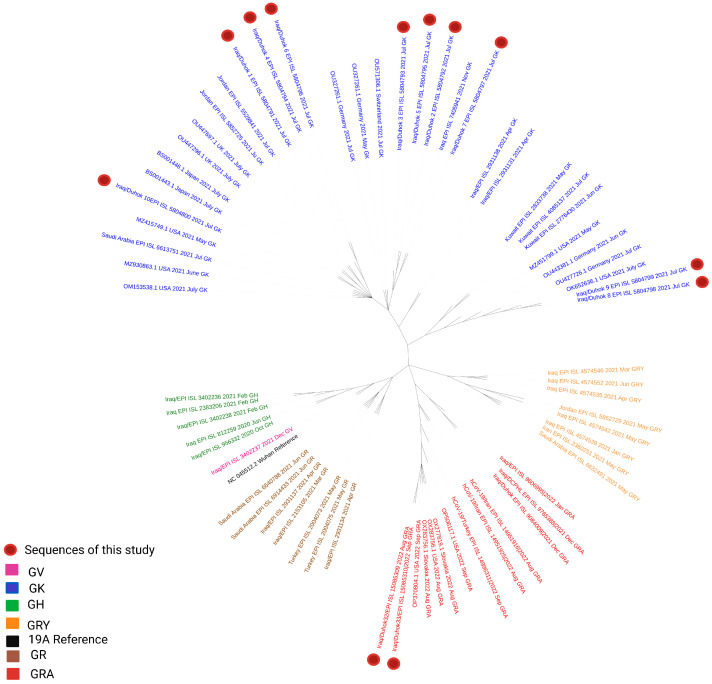
Phylogenetic tree of a dataset including 12 sequences from this study, representative Iraqi isolates, samples from the neighbouring countries, and global sequences.

**Figure 7 diagnostics-13-00008-f007:**
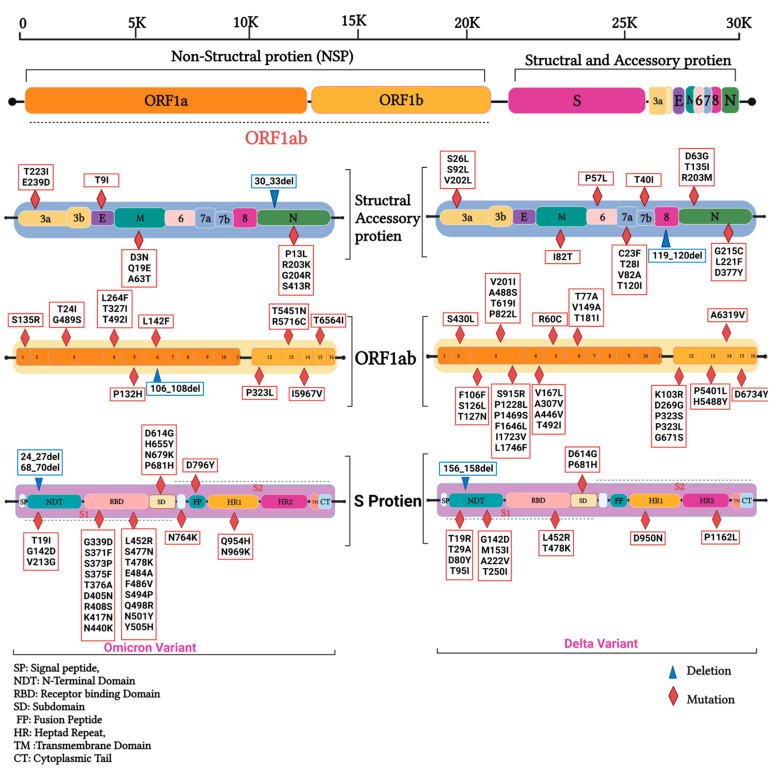
Distribution of amino acid variants on genomes of the Iraqi sequences from the third wave of the SARS-CoV-2 pandemic.

**Figure 8 diagnostics-13-00008-f008:**
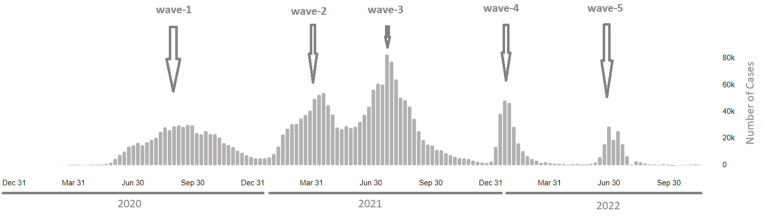
Daily cases and waves of SARS-CoV-2 in Iraq published on the WHO Coronavirus dashboard.

**Table 1 diagnostics-13-00008-t001:** ARS-CoV-2 genetic variations in the isolates from Duhok during the third and fifth waves of the pandemic compared to reference sequence NC_045512.2.

Gene	Nonsynonymous Mutations		Synonymous Mutations		Frameshift Deletion /Non-Frame Deletion		Upstream/Downstream Mutations		Total	
	Delta	Omicron	Delta	Omicron	Delta	Omicron	Delta	Omicron	Delta	Omicron
5’UTR	–	–	–	–	–	–	3	2	3	2
ORF1a	22	8	14	11	–	1	–	–	36	20
ORF1b	10	5	7	2	–	–	–	–	17	7
S	14	29	7	1	1	2	–	–	22	32
ORF3a	3	2	1	2	–	–	–	–	4	4
E	–	1	–	–	–	–	–	–	0	1
M	1	3	1	1	–	–	–	–	2	4
ORF6	1	–	–	–	–	–	–	–	1	0
ORF7a	4	–	–	1	–	–	–	–	4	1
ORF7b	1	–	–	1	–	–	–	–	1	1
ORF8	–	–	–	–	1	–	–	–	1	0
N	6	5	1	1	–	1	–	–	7	7
ORF10	1	–	–	–	–	–	–	–	1	0
3’UTR	–	–	–	–	–	–	3	1	3	1
Total	63 (61.7%)	53(70%)	31 (30.3%)	20 (25%)	2	4	6	3	102	80

## Data Availability

The SARS-CoV-2 sequences of this study are deposited in the Global Initiative on Sharing all Individual Data (GISAID).

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
