# Peer review of "Complete Genomic Characterisation and Mutation Patterns of Iraqi SARS-CoV-2 Isolates"

_diagnostics, 2022, doi:10.3390/diagnostics13010008_

Round 1

Reviewer 1 Report

The authors conducted the complete genome analysis of SARS-CoV-2 during the third and fifth outbreaks in Iraq using 60 samples from the third wave and 2 samples from fifth wave and then combined with 932 strains retrieved from GISAID. I would like to give comments as below. 

1. In introduction, the authors mentioned there was 158 complete sequences from Iraq but in method, the authors collected 932 strains submitted from Iraq at GISAID. Please clarify it. 

2. The sample selection is not clearly described. Although the authors used blind random selection of samples, but there might be a lot of cases during the Omicron outbreak everywhere including the study area and how the authors determined two representative samples from the fifth wave outbreak for this study. 

3. Although the authors discussed about the mutation pattern of Iraqi SARS-CoV-2 strains, if the authors could link to the real time disease outbreak pattern and comparing or exploring the connection between mutation pattern and disease transmission or disease severity or breakthrough infection, or functional changes, could provide the public health interest.

4.  The authors suggested to develop the new diagnostic or vaccine development, please provide the evidence supporting this suggestion. 

5. For more understanding of the readers, please provide information of the study area and the evidence against the representativeness and generalization of the included samples. 

6. Please provide information of the each outbreak episode in Iraq, especially the period interval of each outbreak so that the reader could understand on sampling period with the respective SARS-CoV-2 outbreak period. 

7. The figure quality is poor to evaluate especially those taken from the GISAID or WHO webpage. 

Author Response

Response to Reviewer 1 Comments

The authors conducted the complete genome analysis of SARS-CoV-2 during the third and fifth outbreaks in Iraq using 60 samples from the third wave and 2 samples from fifth wave and then combined with 932 strains retrieved from GISAID. I would like to give comments as below. 

Response: Thanks for all raised comments, regarding the sample number, the samples of this study were 12 and 912 sequences have been retrieved from GISAID, which makes them 924 combined. The number 932 was mistakenly left as the low-quality and short sequences have been excluded from the analysis.

Point 1. In introduction, the authors mentioned there was 158 complete sequences from Iraq but in method, the authors collected 932 strains submitted from Iraq at GISAID. Please clarify it. 

Response 1:       The number of all whole genome sequences submitted to GISAID from Iraq was 924, including sequences of our study (n=12) 924 is the correct number that has been already mentioned in the manuscript in many occasions. The 932 was miss typed as the partial sequences from Iraq were excluded from our analysis.

Point 2. The sample selection is not clearly described. Although the authors used blind random selection of samples, but there might be a lot of cases during the Omicron outbreak everywhere including the study area and how the authors determined two representative samples from the fifth wave outbreak for this study. 

Response 2:  The study area was Duhok Governorate of Iraq. Regarding the Samples from the Fifth wave, however, Sequencing of a large number of samples could be difficult, at Duhok Central public health laboratory, we conduct RT-PCR screening using PowerChek™ SARS-CoV-2 S-gene Mutation Detection Kit ( Kogene)  for some samples for the purpose of variants screening or finding any untypable isolates that could take attention.

Point 3. Although the authors discussed about the mutation pattern of Iraqi SARS-CoV-2 strains, if the authors could link to the real time disease outbreak pattern and comparing or exploring the connection between mutation pattern and disease transmission or disease severity or breakthrough infection, or functional changes, could provide the public health interest.

Response 3: Good point suggested; in this regard the rising of the daily cases and the begining of each wave is clearly linked to the emergence and dominance of a new variant in the community.

Point 4.  The authors suggested to develop the new diagnostic or Vaccine development, please provide the evidence supporting this suggestion. 

Response 4: from the begening of the pandemic, the evolution of SARS-CoV-2 and the emergence of new variants has been a critical issue for many detection systems. Therefore, continuous monitoring studies on the viral evolution and efficacy f the diagnostic methods are required. Regarding the Vaccines, similarly evolutions and mutation of SARS-CoV-2, especially at the immunogenic targeted protein for Vaccine (Spike) has brought about the idea of developing new vaccines containing these newly emerged variants or even thinking about trivalent or a quadrivalent vacines to include more than one variants, on this regard the FDA has approved a bivalent vaccines containing both BA.4 and BA.5 lineages of Omicron variant. Here I have cited two new references supporting our suggestion for Covid-19.

Point 5. For more understanding of the readers, please provide information of the study area and the evidence against the representativeness and generalization of the included samples. 

Response 5: as mentioned in some parts of the study, the study area is Duhok Governorate of Iraq, and the sampling representation was checked before with RT-PCR variants secreening. Therefore we can say that these samples from the third and the fifths waves could be a good represenattives of the variants circulating in the city, supported by the other sequences from other cities of Iraq especially for the third wave sequences. The study area has been added to the sampling section as per your request.

Point 6. Please provide information of the each outbreak episode in Iraq, especially the period interval of each outbreak so that the reader could understand on sampling period with the respective SARS-CoV-2 outbreak period. 

Response 6: the outbreak waves were as follow, the third wave has started at the begening of the June, 2021-Sep, 2021 while the fifths wave was June, 2022-Sep, 2022. These details has been added to the manuscript accordingly.

Point 7. The figure quality is poor to evaluate especially those taken from the GISAID or WHO webpage. 

Response7: the Figures have been updated accordingly.

Reviewer 2 Report

The study provides data regarding the phylogeny of SARS-CoV-2 from an Iraqi region. Despite the small sample size of the study which is a serious drawback, the study is very well written and methodologically well organised. In addition to the analyses performed, I would suggest a network analysis to be carried out in order to depict better the relationships between haplotypes.

Some minor comments:

The authors claim in the results that they performed qPCR as well. However this is not described in the purposes of the study. They should do so explaining why exactly they performed quantification. Did they also use the qPCR results for any other scope or only for control?

In line 109 please provide the primer sequences

In Figure 2, please explain in the Figure legend the GRA, GH, GR etc 

Author Response

Response to Reviewer 2  Comments

The study provides data regarding the phylogeny of SARS-CoV-2 from an Iraqi region. Despite the small sample size of the study which is a serious drawback, the study is very well written and methodologically well organised. In addition to the analyses performed, I would suggest a network analysis to be carried out in order to depict better the relationships between haplotypes.

Response: Thanks for the comments; the sample size is small as Sequencing of a large sample size is expensive, but this drawback has been covered either by muatation and avariant qPCR based detection method and retrieving all other Iraqi isolates to get a representative picture of each pandemic wave in the country.

point 1: The authors claim in the results that they performed qPCR as well. However this is not described in the purposes of the study. They should do so explaining why exactly they performed quantification. Did they also use the qPCR results for any other scope or only for control?

Response 1: The PCR was peformed for the detection purpose and the quantity of the viral RNA for next generation sequencing. The low CTs 9(>20) were selected and  sent for Sequencing.

Point 2: In line 109 please provide the primer sequences

Response 2: The samples have been commercially sequenced by Intergen Genetics and Rare Diseases Diagnosis Research and Application Centre (Ankara, Turkey)

Point 3: In Figure 2, please explain in the Figure legend the GRA, GH, GR etc 

Response 3:These are the GISAID Clades, which are clearly explained in the cited reference.

Round 2

Reviewer 1 Report

The authors responded all the concerns properly. 

In figure 4, could you add the month and year in the horizontal axis.

Figure 6 is so blurred to evaluate. If it is .png (picture format), please consider to use adobe illustrator (ai) or some other tool to improve the quality.

In figure 7, the year is missing and also the column title (for vertical axis). I think it is number of cases. 

Author Response

Response to Reviewer 1 Comments

Response to Reviewer 1 Comments

Point 1: In figure 4, could you add the month and year in the horizontal axis.

Response 1: on the horizantal axis, there is a number of muations. Regarding the years description of mutation emmergence its clearly illustarated in the following Figure (Figure 5).

Point 2: Figure 6 is so blurred to evaluate. If it is .png (picture format), please consider to use adobe illustrator (ai) or some other tool to improve the quality.

Reseponse 2: the figure has been updated with better quality.

Point 3: In figure 7, the year is missing and also the column title (for vertical axis). I think it is number of cases. 

Response 3: The figure has been edited according to the reviewer's request.